# Geranium and Lemon Essential Oils and Their Active Compounds Downregulate Angiotensin-Converting Enzyme 2 (ACE2), a SARS-CoV-2 Spike Receptor-Binding Domain, in Epithelial Cells

**DOI:** 10.3390/plants9060770

**Published:** 2020-06-19

**Authors:** K. J. Senthil Kumar, M. Gokila Vani, Chung-Shuan Wang, Chia-Chi Chen, Yu-Chien Chen, Li-Ping Lu, Ching-Hsiang Huang, Chien-Sing Lai, Sheng-Yang Wang

**Affiliations:** 1Department of Forestry, National Chung Hsing University, Taichung 402, Taiwan; zenkumar@dragon.nchu.edu.tw (K.J.S.K.); mgvani2009@gmail.com (M.G.V.); piscium0312@smail.nchu.edu.tw (C.-S.W.); f8601137@gmail.com (C.-S.L.); 2Bio-Jourdeness International Groups Co. Ltd., Taichung 40462, Taiwan; chiachi@jourdeness.com.tw (C.-C.C.); carriechen@jourdeness.com.tw (Y.-C.C.); lvliping@jourdeness.com.tw (L.-P.L.); huangjingxiang@jourdeness.com.tw (C.-H.H.); 3Agricultural Biotechnology Research Center, Academia Sinica, Taipei 11529, Taiwan

**Keywords:** geranium essential oil, lemon essential oil, citronellol, limonene, ACE2, TMPRSS2, SARS-CoV-2, COVID-19

## Abstract

Severe acute respiratory syndrome coronavirus 2 (SARS-CoV-2), also known as coronavirus disease-2019 (COVID-19), is a pandemic disease that has been declared as modern history’s gravest health emergency worldwide. Until now, no precise treatment modality has been developed. The angiotensin-converting enzyme 2 (ACE2) receptor, a host cell receptor, has been found to play a crucial role in virus cell entry; therefore, ACE2 blockers can be a potential target for anti-viral intervention. In this study, we evaluated the ACE2 inhibitory effects of 10 essential oils. Among them, geranium and lemon oils displayed significant ACE2 inhibitory effects in epithelial cells. In addition, immunoblotting and qPCR analysis also confirmed that geranium and lemon oils possess potent ACE2 inhibitory effects. Furthermore, the gas chromatography-mass spectrometry (GC–MS) analysis displayed 22 compounds in geranium oil and 9 compounds in lemon oil. Citronellol, geraniol, and neryl acetate were the major compounds of geranium oil and limonene that represented major compound of lemon oil. Next, we found that treatment with citronellol and limonene significantly downregulated ACE2 expression in epithelial cells. The results suggest that geranium and lemon essential oils and their derivative compounds are valuable natural anti-viral agents that may contribute to the prevention of the invasion of SARS-CoV-2/COVID-19 into the human body.

## 1. Introduction 

*Coronaviridae* or coronaviruses are a large family of viruses that can cause disease in mammals and birds. In humans, the viruses cause illness, ranging from common cold to severe respiratory diseases. In 2002, severe acute respiratory syndrome (SARS) emerged, and in 2012, Middle East respiratory syndrome (MERS) emerged—both are betacoronoviruses transmitted from animal to human that result in severe respiratory diseases in affected individuals [1]. In December 2019, a novel SARS-coronavirus (CoV)-2, also known as 2019 novel corona virus (2019-nCoV) or coronavirus disease-2019 (COVID-19), caused a pneumonia outbreak in China and subsequently expanded worldwide, leading to a pandemic. Recently, the first genomic sequence of COVID-19 was released, and through comparing to the genomes of SARS-CoV and MERS-CoV, researchers found that COVID-19 has better genomic sequence homology with SARS-CoV than that of MERS-CoV [2,3].

During the first SARS-CoV outbreak, Li et al. [4] identified angiotensin-converting enzyme 2 (ACE2) as the human host factor or cell entry receptor for SARS-CoV. Overexpression of ACE2 and injection of SARS-CoV spike protein developed severe acute lung failure in mice, which can be attenuated by blocking the renin-angiotensin pathway [5]. Recent studies have revealed that COVID-19 spike protein has strong affinity with ACE2 on host cells, which is significantly higher than that of SARS-CoV [1,6]. These studies also pointed out that treatment with transmembrane protease serine 2 (TMPRSS2) inhibitor significantly blocked SARS-CoV cell entry; therefore, either ACE2 or TMPRSS2 blockers can be a potential targets for anti-viral intervention. Another study reported that the COVID-19 receptor binding domain was capable of entering cells expressing human ACE2, while other receptors are ineffective, confirming that human ACE2 is the prime receptor for COVID-19 [7]. Since the host cell receptor plays a crucial role in virus entry, targeting the precise receptor ACE2 is a promising preventive strategy for COVID-19 infection. Recent studies have demonstrated that ACE2 overexpression was frequently observed in gastrointestinal tissues and colon cell lines [1], which is comparatively higher than that of other tissues, including lung tissues. Therefore, in this study, HT-29, a colon adenocarcinoma cell line, was employed to investigate the ACE2 inhibitory effect of test samples in vitro.

At present, there is no definite treatment or vaccine developed for the coronavirus that causes COVID-19. Nevertheless, many possible treatments for COVID-19 have been thrust into the spotlight by scientists and health industries. For example, the anti-malarial drug combination of chloroquine and hydroxychloroquine as well as anti-HIV drugs ritonavir and lopinavir have been recommended [8,9]. Since the drugs directly target the pathogen, the effectiveness of these drugs are largely anecdotal. Additionally, the development of new drugs for targeting ACE2 and treating COVID-19 could be time-consuming. Hence, the safety efficacy of the new drugs are a prime concern, which requires a long time for testing, while the infection is growing fast. The traditional medicine systems from many geographical areas use herbs as the primary treatment of viral infections, including those caused by SARS-CoV. For example, leaf extracts of *Toona sinensis* inhibit SARS-CoV replication [10]. Licorice has been suggested as a promising treatment for SARS-CoV [11]. In addition, natural products including diterpenoids, sesquiterpenoids, triterpenoids, lignoids, curcumin, and ginsenosside-Rb1 have been shown to inhibit SARS-CoV [12,13]. A recent in silico study showed that baicalin, scutellarin, hesperetin, nicotianamine, and glycyrrhizin were capable of inhibiting ACE2 [3]. 

There are a wide-range of essential oils, and their components have been clinically proven to possess antiviral properties [14,15]. A study by Jackwood et al. [16] found that treatment with a mixture of oleoresins and essential oils from botanicals decreased the severity of clinical signs and lesions in chickens that carried the avian infectious bronchitis virus (IBV-CoV). However, the effects of plant essential oils on human coronaviruses are yet to be explored. In particular, their effects on host cell receptors have barely been investigated. Geranium essential oil is derived from the leaves of *Pelargonium graveolens,* which is widely utilized in the cosmetic industry, perfumery, and aromatherapy [17]. Traditionally, geranium essential oil has been used in both physiological and psychological complications, including anxiety, insomnia, high blood pressure, worry, anger, frustration, restlessness, nervousness, weight loss, hypercholesterolemia, gastrointestinal disorders, and respiratory tract infection [18,19]. Geranium oil represents a potent immune modulator, and stimulates and cleans lymphatic system [20]. The oil is clinically used for treating diarrhea, jaundice, diabetes, hepatitis, ulcerative colitis, cholecystitis, and renal stone [21]. The present study aimed to investigate the ACE2 inhibitory effects of plant essential oils and their major components in vitro. 

## 2. Results

### 2.1. Cytotoxic Effect of Essential Oils on HT-29 Cells

Prior to investigation of the ACE2 inhibitory effect of essential oils, we determined the cytotoxicity of essential oils on HT-29 cells by 3-(4,5-dimethyl-thiazol-2-yl)-2,5-diphenyl tetrazolium bromide (MTT) assay. As shown in Table 1, we treated HT-29 cells with increasing concentrations of essential oils (25–200 μg/mL) for 48 h. We found that geranium, cypress, eucalyptus, juniper berry, marjoram, and tea tree oils were not cytotoxic up to a concentration of 200 μg/mL, whereas fennel, frankincense, myrtle, neroli, petitgrain, and tangerine oils displayed moderate cytotoxicity to HT-29 cells, with IC_50_ values of 143.8, 155.91, 144.22, 119.52, 151.63, and 109.31 μg/mL, respectively. Treatment with citronella, may chang, kunzea, palmarosa, black pepper, and elemi oils exhibited strong cytotoxicity towards HT-29 cells, with IC_50_ values of 0.07, 0.11, 0.44, 1.23, 7.62, and 7.93 μg/mL. The histogram analysis of individual essential oils on cell viability is shown in Appendix A. On the basis of the MTT assay, we selected the optimum non-cytotoxic concentration of essential oils for our further ACE2 assay. 

### 2.2. Essential Oils Downregulate ACE2 Activity in HT-29 Cells 

Next, we examined the effect of selected essential oils on ACE2 activity in HT-29 cells. All the treated essential oils exhibited significant inhibition on ACE2 activity (Table 1). Among them, geranium and lemon essential oils strongly reduced the human ACE2 levels in HT-29 cells from 17.68 ng/mL (control) to 1.43 ng/mL and 4.34 ng/mL, respectively (Figure 1A). In addition, we found that geranium and lemon essential oils dose-dependently reduced ACE2 activity in HT-29 cells (Figure 1B,C). Further, to confirm this effect, we examined the protein expression levels of ACE2 by immunoblotting. Similar to that of the ACE2 ELISA assay, the ACE2 protein level was significantly reduced by geranium and lemon essential oils (Figure 2A). Further qPCR analysis also confirmed that geranium and lemon essential oils significantly downregulated ACE2 (Figure 2B) and TMPRSS2 (Figure 2C) mRNA levels in HT-29 cells.

### 2.3. Chemical Compositions of Geranium and Lemon Essential Oils

The major chemical constituents of geranium and lemon oils and their relative amounts were determined by GC–MS analysis. The chemical composition of tested essential oils are listed in Appendix A). The GC–MS profiles and the major compounds of geranium and lemon oils are shown in Figure 3A,B. The relative contents (%) in geranium and lemon essential oils are shown in Table 2 and Table 3, respectively. A total of 22 compounds were identified in geranium oil, accounting for 90.27% of the whole oil. The major component in geranium oils were citronellol (27.1%), geraniol (21.4%), and neryl acetate (10.5 %), which made up around 60% of the content of the whole geranium oil (Table 2). Lemon oil contains nine identified compounds, accounting for 98.1% of the whole oil. The main component of lemon oil was found to be limonene, which represents 73.0% of the whole oil. The second and third most abundant compounds were γ-terpinene (9.2%) and β-pinene (8.6%) (Table 3). 

### 2.4. ACE2 Inhibitory Effects of Major Constituents in Geranium and Lemon Essential Oils

The cytotoxic effect of citronellol, geraniol, limonene, and neryl acetate were determined by MTT assay. Treatment with either citronellol, geraniol, limonene, or neryl acetate did not display cytotoxicity in HT-29 cells up to a concentration of 100 μM for 48 h (Appendix A). Next, we examined the ACE2 inhibitory effects of these compounds. Interestingly, the ACE2 activity was significantly inhibited by citronellol (50 μM), geraniol (50 μM), limonene (50 μM), and neryl acetate (50 μM), as these compound reduced the ACE2 levels from 18.0 ng/mL (control) to 7.67 ng/mL, 10.44 ng/mL, 12.92 ng/mL, and 16.63 ng/mL, respectively (Figure 4A). Similar to the ELISA assay, the immunobloting also confirmed that treatment with citronellol, geraniol, limonene, and neryl acetate significantly inhibited ACE2 protein expression in HT-29 cells (Figure 4B). qPCR analysis further supported this notion that citronellol, geraniol, and limonene significantly downregulate ACE2 mRNA levels, whereas the inhibition of ACE2 mRNA by neryl acetate was not statistically significant (Figure 4C). Interestingly, treatment with major compounds significantly downregulated the TMPRSS2 mRNA levels (Figure 4D), which were involved in S-protein priming during SARS-CoV-2 cell entry. 

## 3. Discussion

Essential oils have been used in folk medicine throughout history as principal ingredients in aromatherapy and psychotherapy. However, modern preclinical studies have revealed that essential oils possess a broad spectrum of pharmacological activities, including antiseptic, diuretic, choleretic, spasmolytic, hyperemic, expectorant, anti-anxiety, antioxidant, anti-inflammatory, and anticancer activities [22]. The pharmacological properties of essential oils are attributed by their unique major constituents, such as monoterpenoids, sesquiterpenoids, and phenylpropanoids. In addition, there is considerable evidence emerging from in vitro studies and controlled trails of the potential for essential oils as antiviral agents for the treatment of human viral infections, including SARS coronaviruses [23]. These antiviral essential oils were mostly tested against enveloped RNA or DNA viruses such as herpes simplex virus type 1, Junin virus, influenza virus, dengue virus type 2, and SARS coronaviruses, as well as non-enveloped viruses such as coxsackie virus B1, poliovirus, and adenovirus type 3 [22]. Most of these clinically useful antiviral agents are substances that act on specific steps of the viral biosynthesis, inhibiting viral replication in particular. On the other hand, virucidal agents denature the structure or glycoproteins of the virus, thus reducing or completely blocking the infectivity of virus particles [24]. 

Recently, ACE2 was identified as a functional SARS coronavirus receptor in epithelial cells. In particular, the SARS-CoV spike protein engages ACE2 as the cell entry receptor and employs the cellular serine protease TMPRSS2 for spike protein priming [1]. Thus, either inhibition of ACE2 or TMPRSS2 receptors can be potential targets for SARS-CoV prevention, as ACE2 displays distinct modes of enzyme action and tissue distribution. Abundant endogenous expression of ACE2 has been observed in cardiovascular and colon-specific cell lines, while fainted expression has been noted in kidney cells [25]. Recent studies have also confirmed that the endogenous expression of ACE2 was higher in gastrointestinal tissues [26] and colon cell lines [1], with these reports pointing out that ACE2 expression is comparatively higher in intestinal tissues than in lung tissues. Therefore, in this study, we employed HT-29, a colon adenocarcinoma cell line, which endogenously expresses ACE2, in order to investigate the ACE2 inhibitory effect of essential oils. Initially, we examined the cytotoxic effects of essential oils in HT-29 cells. Assessment of cytotoxicity is clearly an important aspect of the evaluation of a potent antiviral agent, as a potential agent should be selective for virus- or cell-specific processes with no or limited effects on cellular metabolism [27]. Our results indicated that each essential oil exhibits differential cytotoxicity, some of them are strongly cytotoxic and others of them do not exhibit cytotoxicity up to a concentration of 200 μg/mL. 

Recent in silico studies reported that organosulfur compounds in garlic essential oil and other natural products, such as baicalin, scutellarin, hesperetin, nicotianamine, glycyrrhizin, (*E*,*E*)-α-farnesene, (*E*)-β-farnesene, and (*E*,*E*)−farnesol, have the potential to bind the human ACE2 receptor, thereby possibly blocking SARS-CoV-2 cell entry [3,28,29]. One study evaluated the in vitro antiviral effects against SARS-CoV of seven essential oils from Lebanese species that included *Laurus nobilis* T., *Juniperus oxycedrus* spp. *oxycedrus*, *Thuja orientalis*, *Cupressus sempervirens* spp. *pyramidalis*, *Pistacia palaestina*, *Salvia officinalis*, and *Satureja thymbra*. Among them, the fruit essential oils of *L. nobilis*, *T. orientalis*, and *J. oxycedrus* spp. *oxycedrus* displayed inhibition against SARS-CoV-induced cytopathogenic effect with IC_50_ values of 120 μg/mL, 130 μg/mL, and 270 μg/mL, respectively [27]. However, there are no studies directly exhibiting the ACE2 inhibitory effects of essential oils or their major compounds in vitro or in vivo. In this study, we screened the ACE2 inhibitory effect of 10 essential oils; among them, 8 essential oils exhibited significant ACE2 inhibition. Interestingly, geranium and lemon essential oils strongly inhibited ACE2 activity without displaying cytotoxicity. To the best of our knowledge, this is the first report indicating that geranium and lemon essential oils and their major components citronellol, geraniol, limonene, linalool, and neryl acetate downregulate ACE2 receptor activity in virus–host epithelial cells. 

## 4. Materials and Methods

### 4.1. Chemicals and Reagents

All the essential oils were provided by Bio-Jourdeness International Groups Co. Ltd. (Taichung Taiwan). Citronellol, geraniol, neryl acetate, and limonene were obtained from Tokyo Chemical Industry Co., Ltd. (Tokya, Japan). Purity of these compounds were above 99%, according to the GC and ^1^H-NMR analysis. McCoy’s medium, fetal bovine serum (FBS), sodium pyruvate, penicillin, and streptomycin were obtained from Invitrogen (Carlsbad, CA, USA). 3-(4,5-Dimethyl-thiazol-2-yl)-2,5-diphenyl tetrazolium bromide (MTT) and dimethylsulfoxide (DMSO) were purchased from Sigma-Aldrich. Antibody against ACE2 was obtained from Arigo Biolaboratories (Hsinchu, Taiwan). Antibody against GAPDH was obtained from Cell Signaling Technology Inc. All other chemicals were reagent grade or HPLC grade and were supplied by either Merck or Sigma-Aldrich.

### 4.2. Cell Culture and Cell Viability Assay

Human colorectal adenocarcinoma cell line (HT-29) was cultured and maintained in McCoy’s medium supplemented with 10% fetal bovine serum, 1% L-glutamine, and 1% penicillin/streptomycin. Cell viability was assessed by MTT colorimetric assay. Briefly, HT-29 cells were seeded into a 48-well culture plate with a density of 5 × 10^4^ cells per well. After 24 h incubation, we treated cells with increasing concentrations of essential oils for 48 h. The culture supernatant was removed and 1 mg/mL of MTT in 0.1 mL fresh medium was added. The MTT formazon crystals were dissolved in 0.4 mL of DMSO and the samples were measured at 570 nm (A_570_) using ELISA micro-plate reader (Bio-Tek Instruments, Winooski, VT). The percentage of cell viability (%) was calculated as (A_570_ of treated cells/A_570_ of untreated cells) × 100. 

### 4.3. Determination of ACE2 Activity

Cellular ACE2 activity was measured using a commercially available human ACE2 ELISA kit (Elabscience Biotechnology Inc., Hubei, China). Briefly, HT-29 cells were seeded into a 10 cm cell culture dish with a density of 2 × 10^6^ cells per well. After 48 h, we treated cells with a selective dose (non-cytotoxic high concentration) of essential oils or derived compounds for a further 48 h. Then, the cells were gently washed with pre-cooled phosphate buffered saline (PBS) and we dissociated the cells using trypsin. Cell suspension was centrifuged for 5 min at 1000× *g*. Then, the cells were lysed with a mixture of radioimmuno precipitation assay (RIPA) buffer (Thermo Fisher Scientific, Waltham, MA, USA) and transmembrane protein extraction reagent (Fivephoton Biochemical, San Diego, CA, USA) at a ratio of 1:1, followed by centrifugation for 10 min at 1500× *g* at 4 °C. Protein concentration was determined by Bio-Rad protein assay reagent (Bio-Rad Laboratories, Hercules, CA, USA). Equal amounts of protein samples (100 µg/well) were assayed according to the manufacturer’s protocol. The optical density (OD value) of each sample was determined at 450 nm (A_450_) using an ELISA micro-plate reader (Bio-Tek Instruments).The percentage of ACE2 activity (%) was calculated as (A_450_ of treated cells/A_450_ of untreated cells) × 100.

### 4.4. Determination of ACE2 Protein

HT-29 cells were seeded in a 10 cm dish at a density of 2 × 10^6^ cells per dish. After 24 h, we exposed the cells to test samples for 48 h. Cellular and membrane-bound proteins were isolated by a mixture of RIPA lysis buffer and transmembrane protein extraction reagent. Protein concentration was determined by Bio-Rad protein assay reagent. Equal amounts of denatured protein samples (60 µg) were separated by 10% SDS-PAGE, and the separated proteins were transferred onto polyvinylidene difluoride (PVDF) membrane overnight. Then, the membranes were blocked with 5% non-fat dried milk for 30 min, followed by incubation with ACE2 or GAPDH antibodies overnight, and then incubated with either horseradish peroxidase-conjugated goat anti-rabbit or anti-mouse antibodies for 1 h. Immunoblots were developed with the enhanced chemiluminescence (ECL) reagents (Advansta Inc., San Jose, CA, USA), images were captured by CheniDoc XRS^+^ docking system, and the protein bands were quantified by using Imagelab software (Bio-Rad laboratories, Hercules, CA, USA). 

### 4.5. Quantitative Real-Time PCR

Total RNA was extracted by total RNA purification kit (GeneMark, New Taipei City, Taiwan). RNA concentration was quantified with a NanoVue Plus spectrophotometer (GE Health Care Life Sciences, Chicago, IL, USA). Quantitative PCR (qPCR) was performed on a real-time PCR detection system and software (Applied Biosystems, Foster City, CA, USA). First-strand complementary DNA (cDNA) was generated by SuperScript III reverse transcriptase kit (Invitrogen). Quantification of mRNA expression for genes of interest was performed by qPCR reactions with equal volume of cDNA, forward and reverse primers (10 µM), and power SYBR Green Master Mix (Applied Biosystems, Foster city, CA, USA). The sequence of the PCR primers were as follows: ACE2: forward 5′-GCTGCTCAGTCCACCATTGAG-3′, reverse 5′-GCTTCGTGGTTAAACTTGTCCAA-3′; TMPRSS2: forward 5′-AATCGGTGTGTTCGCCTCTAC-3′, reverse 5′-GCGGCTGTCACGATCC-3′; GAPDH: forward 5′-TCCTGGTATGACAACGAAT-3′, reverse 5′-GGTCTCTCTCTTCCTCTTG-3′ [30]. The copy number of each transcript was calculated as the relative copy number normalized by GAPDH copy number.

### 4.6. GC–MS Analysis

To determine the essential oil composition, we carried out analyses using an ITQ 900 mass spectrometer coupled with a DB-5MS column. The temperature program was as follows: 45 °C for 3 min, then increased to 3 °C/min to 180 °C, and then increased to 10 °C/min to 280 °C hold for 5 min. The other parameters were injection temperature, 240 °C; ion source temperature, 200 °C; EI, 70 eV; carrier gas, He 1 mL/min; and mass scan range, 40–600 *m/z*. The volatile compounds were identified by Wiley/NBS Registry of mass spectral databases (V. 8.0, Hoboken, NJ, USA), National Institute of Standards and Technology (NIST) Ver. 2.0 GC/MS libraries and the Kovats indices were calculated for all volatile constituents using a homologous series of *n*-alkanes C_9_-C_24_. The major components were identified by co-injection with standards (wherever possible).

### 4.7. Statistical Analysis

Data are expressed as mean ± SD. All data were analyzed using the statistical software Graphpad Prism version 6.0 for Windows (GraphPad Software, San Diego, CA, USA). Statistical analysis was performed using one-way ANOVA followed by Dunnett’s test for multiple comparison. *p*-values of less than 0.05 *, 0.01 **, 0.001 ***, and 0.0001 **** were considered statistically significant for the sample treatment group vs. the control group. 

## 5. Conclusions 

The recent COVID-19/SARS-CoV-2 pneumonia pandemic constitutes the largest global public health crisis and has created international anxiety due to its relatively high infectious, rapid progression, as well as its relatively high death rate. Drug development for treating COVID-19 is currently important due to its rapid progression. The conventional antiviral vaccine development is time-consuming and its safety needs to be verified. The fact is that there is no conventional medicine developed for the treatment of COVID-19. However, there is evidence that suggests that essential oils and their major components have displayed potent antiviral activity to other coronaviruses, such as SARS-CoV-1, although the mechanism of action of these oils and their components were found to be mainly through inhibition of viral replication [27]. Recently, ACE2, a receptor in host cell favoring virus cell entry, was recognized as one of the prime targets to minimize the infection. In this study, we presented the first piece of evidence that geranium and lemon essential oils and their major compounds, citronellol, geraniol, limonene, linalool, and neryl acetate, could downregulate ACE2 expression in epithelial cells, thereby blocking virus entry into host cells, and eventually preventing viral infection. However, further studies are highly warranted to unveil the underlying molecular mechanisms of this inhibitory effect.

## Figures and Tables

**Figure 1 plants-09-00770-f001:**
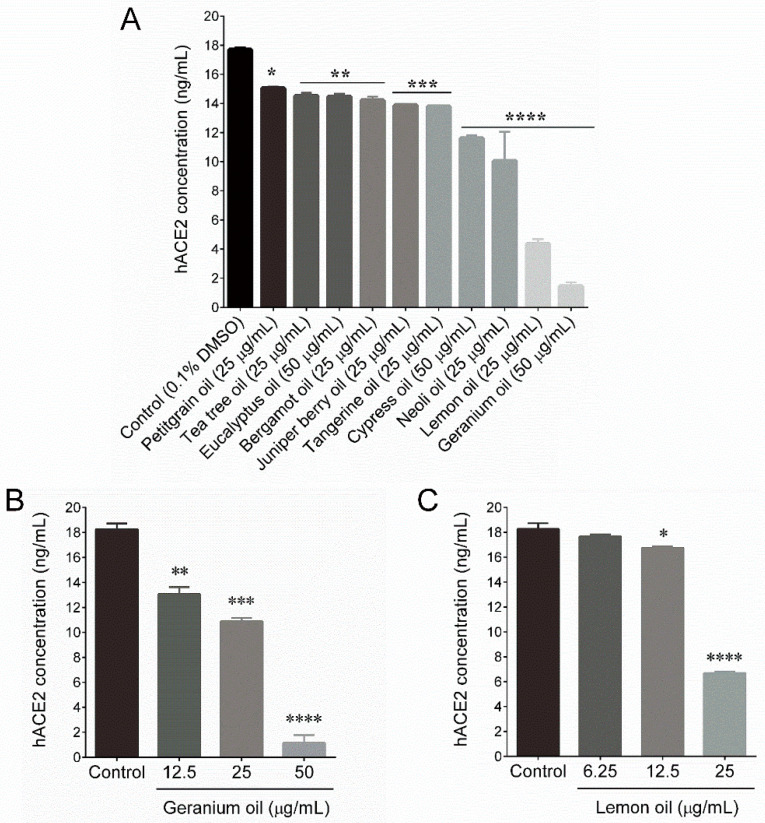
Essential oils reduced human angiotensin-converting enzyme 2 (ACE2) levels in HT-29 cells. (**A**) The ACE2 inhibitory effects of selected essential oils. HT-29 cells were incubated with indicated concentration of selected essential oils for 48 h. Cell lysates were subjected to determine ACE2 levels using a commercially available ELISA kit. (**B**,**C**) The dose-dependent inhibitory effects of geranium and lemon essential oils were determined in the same course of time. Values represent the mean ± SD of three independent experiments. *p*-values of less than 0.05 *, 0.01 **, 0.001 ***, and 0.0001 **** were considered statistically significant for the sample treatment group vs. the control group.

**Figure 2 plants-09-00770-f002:**
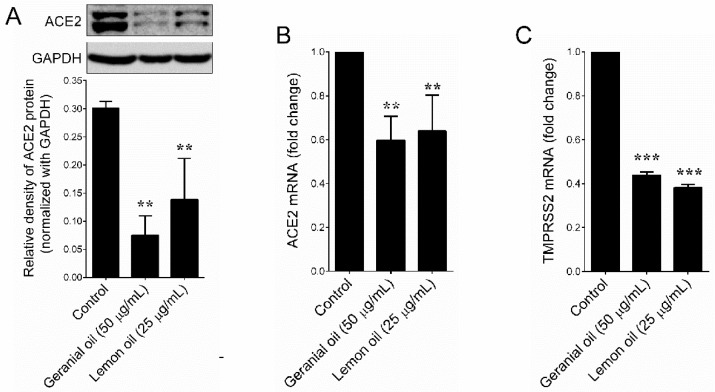
Geranium and lemon essential oils downregulated ACE2 expression in HT-29 cells. (**A**) Protein expression levels of ACE2 were determined by immunoblotting. Glyceraldehyde 3-phosphate dehydrogenase (GAPDH) was used as an internal loading control. Relative density of one representative experiment is shown, where ACE2 signal was normalized to GAPDH signal. (**B**,**C**) Relative expression of ACE2 and TMPRSS2 mRNAs in HT-29 cells. Total RNA was extracted from cells treated with indicated concentration of geranium and lemon essential oils for 48 h. The transcription levels of ACE2 and TMPRSS2 were quantified by qPCR and a representative experiment is shown. The Δ^ct^ values of ACE2 and TMPRSS2 mRNAs were normalized to GAPDH mRNA. Values represent the mean ± SD of three independent experiments. *p*-values of less than 0.01 **, and 0.001 *** were considered statistically significant for the sample treatment group vs. the control group.

**Figure 3 plants-09-00770-f003:**
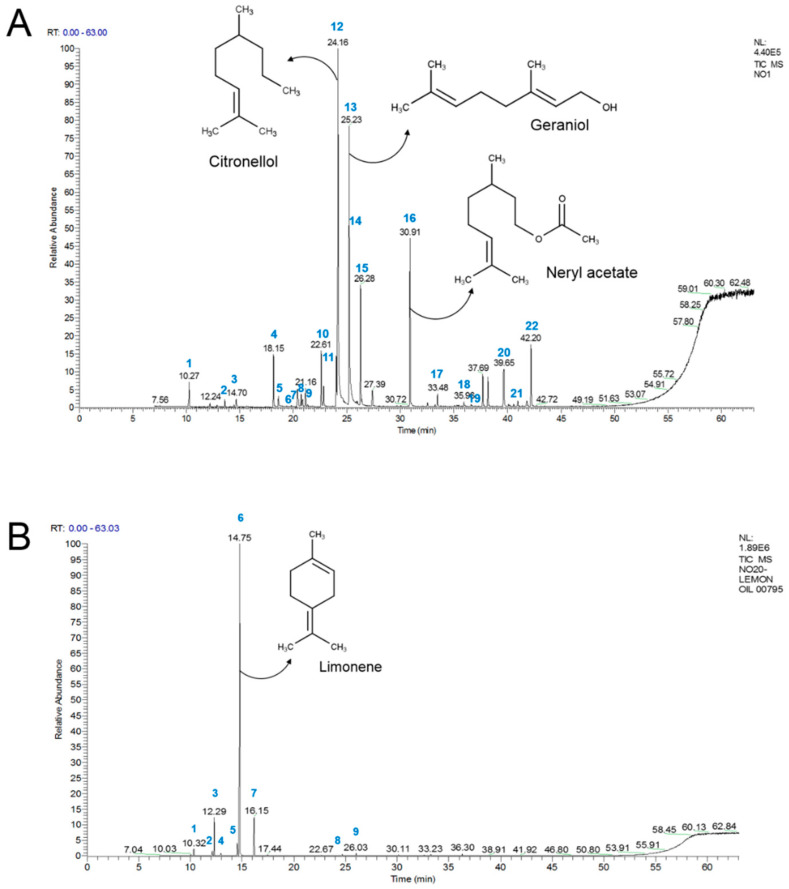
GC–MS analysis of geranium and lemon essential oils. GC profiles and chemical structures of the major compounds in geranium (**A**) and lemon (**B**) essential oils.

**Figure 4 plants-09-00770-f004:**
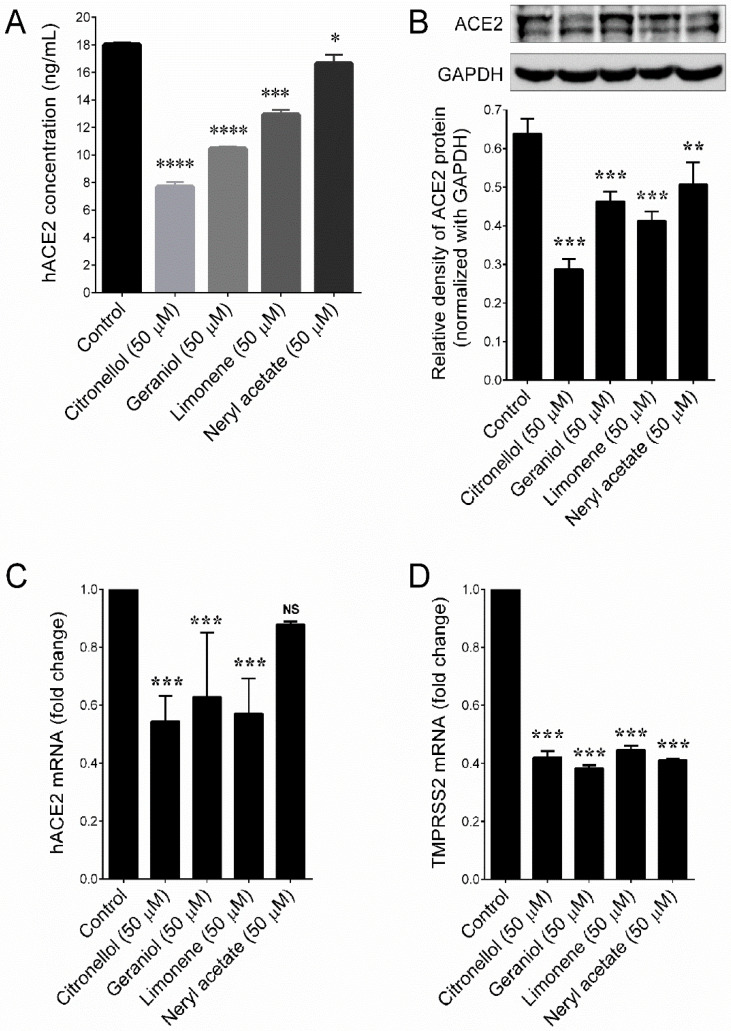
ACE2 inhibitory effects of major compounds of geranium and lemon essential oils. (**A**) ACE2 activity in HT-29 cells were determined by ELISA assay. (**B**) Protein expression levels of ACE2 were determined by immunoblotting. Relative density of one representative experiment is shown, where ACE2 signal was normalized to GAPDH signal. (**C**,**D**) Relative expression of ACE2 and TMPRSS2 mRNAs in HT-29 cells. Values represent the mean ± SD of three independent experiments. *p*-values of less than 0.05 *, 0.01 **, 0.001 ***, and 0.0001 **** were considered statistically significant for the sample treatment group vs. the control group.

**Table 1 plants-09-00770-t001:** Cytotoxic and ACE2 inhibitory effects of essential oils.

No	Essential Oil	Botanical Name	IC_50_ (μg/mL)	Selected Dose (μg/mL)	ACE2 Activity (% of Control)
1	Bergamot oil	*Citrus bergamia*	48.58	25	66.54 ^***^
2	Black pepper oil	*Piper nigrum*	<25	-	-
3	Chamomile German blue oil	*Matricaria chamomilla*	<25	-	-
4	Chili oil	*Capsicum annum*	>200	-	-
5	Citronella oil	*Cymbopogon winterianus*	<25	-	-
6	Clary sage oil	*Salvia sclarea*	39.22	-	-
7	Cypress oil	*Cupressus sempervirens*	>200	50	65.12 ^***^
8	Elemi oil	*Canarium valgare*	<25	-	-
9	Eucalyptus oil	*Eucalyptus globulus*	>200	50	70.50 ^***^
10	Fennel oil	*Foeniculum vulgare*	97.22	25	87.3 ^*^
11	Frankincense oil	*Boswellia* sp.	155.91	25	91.48
12	Geranium oil	*Pelargonium graveolens*	>200	50	10.63 ^****^
13	Ginger oil	*Zingiber officinale*	83.03	25	79.52 ^**^
14	Juniper berry oil	*Juniperus communis*	>200	50	60.85 ^***^
15	Kunzea oil	*Kunzea ambigua*	<25	-	-
16	Lemon oil	*Citrus limon*	57.93	25	24.79 ^****^
17	Lavender oil	*Lavandula officinalis*	55.65	-	-
18	Lime oil	*Citrus aurantifolia*	50.98	-	-
19	May chang oil	*Litsea cubeba*	<25	-	-
20	Marjoram oil	*Origanum majorana*	>200	50	90.11
21	Myrtle oil	*Myrtus communis*	144.22	25	93.56
22	Neroli oil	*Citrus aurantium*	119.52	25	52.88 ^***^
23	Palmarosa oil	*Cymbopogon martinii*	<25	-	-
24	Patchouly oil	*Pogostemon cablin*	22.19	-	-
25	Peppermint oil	*Mentha piperita*	40.96	-	-
26	Petitgrain oil	*Citrus aurantium*	151.63	25	80.23 ^**^
27	Ravinstra oil	*Cinnamomum camphora*	45.58	-	-
28	Rosemary oil	*Rosmarinus officinalis*	32.44	-	-
29	Tangerine oil	*Citrus reticulata*	109.31	25	59.20 ^***^
30	Tea tree oil	*Mellaleuca alternifolia*	>200	50	71.55 ^***^

Values represent the mean ± SD of three independent experiments. *p*-values of less than 0.05 *, 0.01 **, 0.001 ***, and 0.0001 **** were considered statistically significant for the sample treatment group vs. the control group.

**Table 2 plants-09-00770-t002:** The main components and their relative contents (%) of geranium oil.

No	Compound	RT (min)	Contents (%)	KI	Identification *
1	α-Pinene	10.27	1.2	934	MS, KI, ST
2	α-Myrcene	13.6	0.4	1004	MS, KI, ST
3	Limonene	14.7	0.5	1029	MS, KI, ST
4	Linalool	18.5	0.87	1098	MS, KI, ST
5	Phenylethyl alcohol	18.62	0.8	1109	MS, KI, ST
6	Isopulegol	20.4	1.6	1148	MS, KI, ST
7	Menthone	20.74	0.7	1155	MS, KI, ST
8	Citronellal	20.87	0.4	1158	MS, KI, ST
9	iso-Menthone	21.16	1.1	1164	MS, KI, ST
10	α-Terpineol	22.61	3.6	1192	MS, KI, ST
11	Nerol	24.03	3.3	1124	MS, KI, ST
12	Citronellol	24.16	27.1	1153	MS, KI, ST
13	Geraniol	25.23	21.4	1251	MS, KI, ST
14	Neral	25.98	0.2	1267	MS, KI, ST
15	Citronellyl formate	26.28	7.7	1273	MS, KI, ST
16	Neryl acetate	30.91	10.5	1377	MS, KI, ST
17	Aristoene	33.48	0.8	1439	MS, KI, ST
18	Germacrene D	35.96	0.3	1375	MS, KI
19	δ-Cadinen	35.61	0.2	1514	MS, KI
20	Guaiol	39.65	2.7	1592	MS, KI, ST
21	Eudesmol	41.8	0.6	1650	MS, KI, ST
22	α-Bisabolol	42.2	4.3	1661	MS, KI

RT: retention time; * MS: National Institute of Standards and Technology (NIST) and Wiley libraries and the literature; KI: Kovats index on a DB-5MS column in reference to *n*-alkanes; ST: authentic standard compounds.

**Table 3 plants-09-00770-t003:** Main components and their relative contents (%) of lemon oil.

No	Compound	RT (min)	Contents (%)	KI	Identification *
1	α-Pinene	10.32	1.4%	935	MS, KI, ST
2	Sabinene	12.11	1.2%	974	MS, KI, ST
3	β-Pinene	12.29	8.6%	978	MS, KI, ST
4	β-Myrcene	12.94	0.6%	991	MS, KI, ST
5	*p*-Cymene	14.52	3.0%	1025	MS, KI, ST
6	Limonene	14.75	73.0%	1030	MS, KI, ST
7	γ-Terpinene	16.15	9.2%	1060	MS, KI, ST
8	Neral	24.66	0.5%	1238	MS, KI, ST
9	Geranial	26.03	0.7%	1268	MS, KI, ST

RT: retention time; * MS: NIST and Wiley libraries and the literature; KI: Kovats index on a DB-5MS column in reference to *n*-alkanes; ST: authentic standard compounds.

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
