# Peer review of "Geranium and Lemon Essential Oils and Their Active Compounds Downregulate Angiotensin-Converting Enzyme 2 (ACE2), a SARS-CoV-2 Spike Receptor-Binding Domain, in Epithelial Cells"

_plants, 2020, doi:10.3390/plants9060770_

Round 1

Reviewer 1 Report

In this manuscript the authors screened ACE2 inhibitors from a pool of plant essential oils utilizing cell based assay, and reported that geranium and lemon essential oils and their derived pure compounds were potent ACE2 inhibitors. 

There are many reports indicate that plant essential oils possessed anti-viral properties through directly suppressing viral growth or replication. However, this study is quite interesting from the standpoint of preventing viral entry into the host cells. Without doubt, this study may provide a new insight for the prevention of human coronavirus infection, particularly the recent COVID-19 by utilizing plant volatiles and their bioactive compounds. 

I suggest to accept this manuscript for the publication in the journal, PLANTS. However, few minor correction has to be made before publication. As follows, 

There are few typo errors were found throughout the manuscript such as., Line 20, oils; line 39, CoV; line 69, anti-viral; line 128, GAPDH; line 138, neryl acetate; figure 4a, citronellol; line 218, CoV; line 235, geranial; line 235, linalool; line 263, 1500 or 15000 g?

In introduction, the authors provided a background information only for geranium oil. Therefore, I suggest to briefly include background studies of lemon oil, especially their anti-viral properties and mechanisms if possible.

Author Response

Reviewer: 1

Reviewer’s opinion: In this manuscript the authors screened ACE2 inhibitors from a pool of plant essential oils utilizing cell based assay, and reported that geranium and lemon essential oils and their derived pure compounds were potent ACE2 inhibitors. There are many reports indicate that plant essential oils possessed anti-viral properties through directly suppressing viral growth or replication. However, this study is quite interesting from the standpoint of preventing viral entry into the host cells. Without doubt, this study may provide a new insight for the prevention of human coronavirus infection, particularly the recent COVID-19 by utilizing plant volatiles and their bioactive compounds. I suggest to accept this manuscript for the publication in the journal, PLANTS. However, few minor correction has to be made before publication.

Response:  We would appreciate the reviewer-1, whom provided a positive comments on our work. The reviewer’s comments and our responses as follows.

Comment. 1: There are few typo errors were found throughout the manuscript such as., Line 20, oils; line 39, CoV; line 69, anti-viral; line 128, GAPDH; line 138, neryl acetate; figure 4a, citronellol; line 218, CoV; line 235, geranial; line 235, linalool; line 263, 1500 or 15000 g?.

Response: We thank the reviewer for pointing out these typographical errors. The indicated mistake were corrected.

Page 1, Line 25. “oils” was replaced with “oil”.

Page 2, Line 45. “Cov” was replaced with “CoV”.

Page 2, Line 75. “anti-viral” was corrected as “antiviral”.

Page 5, Line 141. “GPDH” was corrected as “GAPDH”.

Page 5, Line 152. “nerol” was corrected as “neryl”.

Figure 4a, the “citrunellol” was corrected as “citronellol”.

Page 9, Line 234. “Cov” was corrected as “CoV”.

Page 10, Line 255. “geranoil” was corrected as “geranial”.

Page 10, Line 255. “Linalool” was replaced by “neryl acetate”.

Page 10, Line 283. We double checked. It is 1500 ´ g.

Comment. 2: In introduction, the authors provided a background information only for geranium oil. Therefore, I suggest to briefly include background studies of lemon oil, especially their anti-viral properties and mechanisms if possible.

Response:  There was no such anti-viral effect of lemon essential oil was reported. Whereas, antibacterial effects of lemon oils were reported. Thus, we does not included the antiviral effects of lemon oil in the introduction.

Reviewer 2 Report

The authors presented the ACE2 inhibitory effect of plant essential oils and their major components. As ACE2 blockers can be a potential target for anti-viral intervention, especially as COVID-19 spike protein has strong affinity with ACE2 on host cells. The analysis was performed by GC-MS technique and was done correctly. Generally, this study represents highly interested study for the readers, but before acceptance some of the issues given below should be considered and corrected:

  1. Introduction

Page 2, Table 1 should be under 2.1 and not in the Introduction. Use Latin names for the oils. As these oils were obtained commercially, it should contain declaration of their major constituents. This information would be useful for the readers when considering the activities tested. Please add this information in the supplement. E.g. bergamot oil and lime oil major constituent is also limonene as in the tested lemon oil.

  1. Results

Line 94  To test the cytotoxic of essential oils on HT-29 cells, concentration from 25 to 200 µg/mL were used. For many essential oils (no. 2, 3, 5, 8, 15, 17, 23) the IC50 given in Table 1 are far below these concentrations and should be reevaluated using at least three concentrations to evaluate IC50 or change to ˂25 µg/mL.

Please re-check IC50 of Chilli oil and Fennel oil as according to the supplement chilli oil should be over 100 µg/mL (instead 47.75 µg/mL) and fennel oil seems between 50 and 100 µg/mL (instead 143.8 µg/mL). In addition, please use the same designation in Table1 and Supplement for lavender and lime oil (with or without “organic”).

Page 6   Figure 3A. Please correct citronellol structure, i.e. do not show enantiomer (only general structure) as the other one also exist in nature – chiral column was not used!

Page 8, line 208 “…200 µg/mL.”

Author Response

Reviewer: 2

Reviewer’s opinion: The authors presented the ACE2 inhibitory effect of plant essential oils and their major components. As ACE2 blockers can be a potential target for anti-viral intervention, especially as COVID-19 spike protein has strong affinity with ACE2 on host cells. The analysis was performed by GC-MS technique and was done correctly. Generally, this study represents highly interested study for the readers, but before acceptance some of the issues given below should be considered and corrected:

Response:  We thank the reviewer-2 provided such a positive and constructive comments on our work.  The responses to reviewer’s comments as follows.

Comment. 1: Introduction. Page 2, Table 1 should be under 2.1 and not in the Introduction. Use Latin names for the oils. As these oils were obtained commercially, it should contain declaration of their major constituents. This information would be useful for the readers when considering the activities tested. Please add this information in the supplement. E.g. bergamot oil and lime oil major constituent is also limonene as in the tested lemon oil.

Response: Thank you for pointing it out.

As per your suggestion, the Table. 1 was moved under sub-section 2.1 (Page 4, Line 120).

As per your suggestion, the Latin names of essential oils were included in Table. 1.

As per your suggestion, the declaration of their major constituents of essential oils were provided in the supplementary information section (Table S1~ S8), also mentioned in the text (Page 5, Line 147).

Comment. 2: Results. Line 94 to test the cytotoxic of essential oils on HT-29 cells, concentration from 25 to 200 µg/mL were used. For many essential oils (no. 2, 3, 5, 8, 15, 17, 23) the IC50 given in Table 1 are far below these concentrations and should be reevaluated using at least three concentrations to evaluate IC50 or change to ˂25 µg/mL.

Response:

Thank you for pointing out these mistakes. As per your suggestion, we replaced the IC50 values with <25 mg/mL in which essential oils exhibited below 25 mg/mL (Table. 1).

Comment. 3: Please re-check IC50 of Chilli oil and Fennel oil as according to the supplement chilli oil should be over 100 µg/mL (instead 47.75 µg/mL) and fennel oil seems between 50 and 100 µg/mL (instead 143.8 µg/mL). In addition, please use the same designation in Table1 and Supplement for lavender and lime oil (with or without “organic”).

Response:

Thank you for pointing it out.

As per your suggestion, the IC50 values of chilli and fennel oils were recalculated and represented in Table. 1.

In addition, the designation of lime oil and lavender oils were corrected in both Table. 1 and supplementary Figure 1.

Comment. 5: T Page 6   Figure 3A. Please correct citronellol structure, i.e. do not show enantiomer (only general structure) as the other one also exist in nature – chiral column was not used!

Response: Thank you for pointing out this. As per your suggestion, the enantiomer in citronellol’s structure was removed and showed only the general structure (Figure 3A).

Comment. 6: Page 8, line 208 “…200 µg/mL.”

Response: Thank you for pointing this out. We corrected the above indicated typos (Page 9, Line 230).